# Enhancing Robustness in Learning with Noisy Labels: An Asymmetric Co-Training Approach

## ABSTRACT

Label noise, an inevitable issue in various real-world datasets, tends to impair the performance of deep neural networks. A large body of literature focuses on symmetric co-training, aiming to enhance model robustness by exploiting interactions between models with distinct capabilities. However, the symmetric training processes employed in existing methods often culminate in model consensus, diminishing their efficacy in handling noisy labels. To this end, we propose an **A**symmetric **C**o-**T**raining (**ACT**) method to mitigate the detrimental effects of label noise. Specifically, we introduce an asymmetric training framework in which one model (*i.e.*, RTM) is robustly trained with a selected subset of clean samples while the other (*i.e.*, NTM) is conventionally trained using the entire training set. We propose two novel criteria based on agreement and discrepancy between models, establishing asymmetric sample selection and mining. Moreover, a metric, derived from the divergence between models, is devised to quantify label memorization, guiding our method in determining the optimal stopping point for sample mining. Finally, we propose to dynamically re-weight identified clean samples according to their reliability inferred from historical information. We additionally employ consistency regularization to achieve further performance improvement. Extensive experimental results on synthetic and real-world datasets demonstrate the effectiveness and superiority of our method. The source code has been made anonymously available at https://github.com/shtdusb/ACT.

## CCS CONCEPTS

• **Computing methodologies → Machine learning**.

## KEYWORDS

Noisy labels, asymmetric co-training, sample selection

## 1 INTRODUCTION

Deep neural networks (DNNs) are renowned for their remarkable effectiveness in various computer vision tasks, including image classification [27], object detection [33], face recognition [5], and instance segmentation [60]. Among all factors contributing to the efficacy of deep neural networks, the availability of large-scale, high-quality human-labeled training data [8] is recognized as instrumental in ensuring their state-of-the-art (SOTA) performances. However, such

Permission to make digital or hard copies of all or part of this work for personal or classroom use is granted without fee provided that copies are not made or distributed for profit or commercial advantage and that copies bear this notice and the full citation on the first page. Copyrights for components of this work owned by others than the author(s) must be honored. Abstracting with credit is permitted. To copy otherwise, or republish, to post on servers or to redistribute to lists, requires prior specific permission and/or a fee. Request permissions from permissions@acm.org.
*ACM MM, 2024, Melbourne, Australia*
© 2024 Copyright held by the owner/author(s). Publication rights licensed to ACM.
ACM ISBN 978-x-xxxx-xxxx-x/YY/MM
https://doi.org/10.1145/nnnnnnn.nnnnnnn

large volumes of accurate human annotations are costly and time-consuming to acquire, especially for tasks that necessitate expert annotating knowledge (*e.g.*, medical images [49]). To obtain large-scale annotated data under a limited budget, recent researchers have started to pay attention to using crowd-sourcing platforms [48] or web image search engines [10] for dataset construction. Unfortunately, these methods inevitably introduce low-quality samples with noisy labels, which can cause DNNs to overfit misleading information and degrade their performance. Consequently, developing robust methods aimed at alleviating the detrimental impact of noisy labels is of significant importance.

Prior literature has illustrated the **_Memorization Effect_** [2] of DNNs, which suggests that models tend to first learn clean samples and then progressively memorize noisy ones. Accordingly, researchers explore a diversity of robust learning strategies, such as sample selection [6, 7, 22, 50, 53, 55], label correction [41, 58, 59] and sample re-weighting [9, 21, 44], to mitigate the harmful effects of noisy labels. Notably, among existing solutions, the symmetric co-training (SCT) is one of the most popular training strategies within the realm of sample-selection methods [13, 28, 38, 40, 45, 61].

SCT methods usually entail the simultaneous training of two networks with identical architectures but distinct weight initialization. The twin networks adopt the same training strategy, capitalizing on their distinct learning capabilities to provide mutual guidance throughout the learning process, as shown in Fig. 1 (a). For example, Decoupling [28] trains two networks simultaneously and updates them using instances with different predictions. Co-teaching [13] maintains two networks simultaneously and enables them to select low-loss samples for each other. Co-teaching+ [61] follows a similar scheme as Co-teaching but proposes to select small-loss data from disagreement one. JoCoR [45] employs a joint loss to select low-loss data, encouraging agreement between networks. The efficacy of SCT methods primarily relies on the assumption that the two networks can extract divergent knowledge from the training data, thereby augmenting robustness through complementary information. However, we argue that the information gains attributed to SCT are substantially constrained since the capability discrepancies between the twin networks mainly arise from distinct initializations. Furthermore, it is problematic that the learning capabilities of the twin networks tend to converge in the later stage of training, leading to a decline in effectiveness for addressing noisy labels [40].

To alleviate aforementioned issues, we propose a novel approach, termed **ACT** (**A**symmetric **C**o-**T**raining), to combat noisy labels, as shown in Fig. 1 (b). In our ACT approach, two models with identical architectures are simultaneously trained utilizing distinct training strategies. The first model, designated as the Robustly Trained Model (RTM), is trained with a selected clean subset. Contrarily, the second model, termed the Non-Robustly Trained Model (NTM), undergoes training on the entire noisy training set. Owing to our asymmetric training strategy, we empower the robustness of the

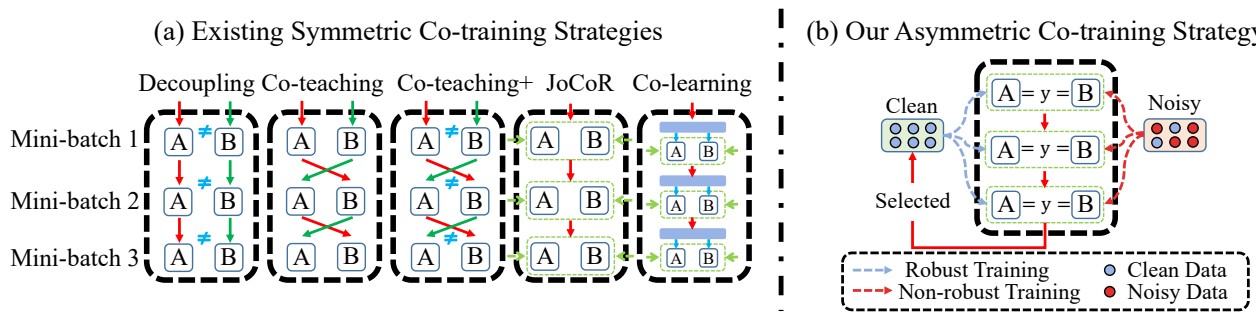

**Figure 1: The differences between classic symmetric co-training methods (*i.e.*, Decoupling, Co-teaching, Co-teaching+, JoCoR and Co-learning) and our asymmetric co-training approach.**

RTM by capitalizing on the diverse capabilities of the two models. In our framework, we introduce two novel criteria to devise an asymmetric sample selection and mining strategy that hinges on the relationship between model predictions and given labels, focusing on both consensus and disagreement. Moreover, we propose a dynamic sample re-weighting approach to leverage the historical states throughout the training process, enhancing the reliability of our clean sample selection and mining. Employing two asymmetrically trained models, our ACT establishes a positive feedback loop that continuously promotes the model's robustness against noisy labels. Notably, this enhancement is achieved without the requirement for any dataset-dependent prior knowledge (*e.g.*, a pre-defined noise rate and a small subset of clean samples). Comprehensive experimental results have been provided to verify the effectiveness and superiority of our approach. Our main contributions are summarized as follows:

(1) We propose a novel asymmetric co-training (ACT) approach to mitigate the negative impact induced by noisy labels. It trains two networks asymmetrically to improve the reliability of learned knowledge. Through this asymmetric training framework, our RTM and NTM can provide more distinctive insights for clean sample selection compared to existing SCT methods.

(2) We introduce two novel criteria to establish an asymmetric sample selection and mining strategy based on the relationship between model predictions, focusing on their consensus and disagreement with given labels. Moreover, we propose a dynamic sample re-weighting method, utilizing historical training states to enhance the reliability of our clean sample selection and mining.

(3) We present comprehensive experimental results on both synthetic and real-world datasets to demonstrate the superiority of our proposed ACT method. Moreover, we conduct extensive ablation studies to further validate the effectiveness of our approach.

## 2 RELATED WORK

### 2.1 Learning with Noisy Labels
Researchers have explored various robust training strategies for learning with noisy labels (LNL) [3, 6, 7, 15, 19, 24, 25, 29, 54, 65]. Existing LNL methods can be categorized into three main directions: sample selection [13, 21, 23, 51], label correction [1, 6, 11, 21, 52, 58], and sample re-weighting [9, 34, 43, 44, 46, 63].

**Sample Selection**: To cope with noisy labels, one intuitive idea is to select clean samples and discard noisy ones from training [23, 53, 56]. Previous sample selection methods primarily regard samples with small losses as clean ones [13, 28, 45, 61]. For instance, DivideMix [21] extracts the clean subset by fitting the loss distribution with the Gaussian Mixture Model. Some recent methods propose new selection criteria for finding clean samples [20, 30]. For example, NCE [20] resorts to neighbor data to identify clean and noisy samples. BARE [30] proposes a data-dependent, adaptive sample selection strategy that relies on batch statistics of a given mini-batch. However, these methods usually demand pre-defined drop rates or thresholds to facilitate efficient selection.

**Label Correction**: Another straightforward idea for addressing noisy labels is to correct corrupted labels before feeding them into networks [1, 6, 11, 12, 31, 52]. Label correction methods typically attempt to rectify sample labels using the noise transition matrix [11] or model predictions [21]. For example, Goldberger *et al.* [11] proposes to use an additional layer to estimate the noise transition matrix. Jo-SRC [56] uses the temporally averaged model (*i.e.*, mean-teacher model) to generate reliable pseudo-label distributions for providing supervision. However, the noise transition matrix is difficult to estimate accurately, while prediction-based label correction tends to suffer from error accumulation.

**Sample Re-weighting**: Recently, some researchers have focused on re-weighting training samples to cope with noisy labels [9, 34, 36, 42, 44]. For example, DIW [9] proposes a dynamic importance weighting strategy as an end-to-end solution to alleviate the bias of static importance weighting. RPM [44] proposes a Bayesian method that infers the example weights as latent variables. L2RW [34] proposes to assign different sample weights based on meta-learning. However, existing sample re-weighting methods also tend to require dataset-dependent prior knowledge (*e.g.*, a small subset of clean samples), posing a limit to their practicability.

### 2.2 Symmetric Co-training
Symmetric co-training is one of the most frequently-employed strategies in sample selection methods [13, 21, 28, 38, 40, 45, 56, 61]. The idea of SCT stems from the Co-training approach [4], which aims to obtain information gains by simultaneously training two models and enabling them to mutually guide the learning process.

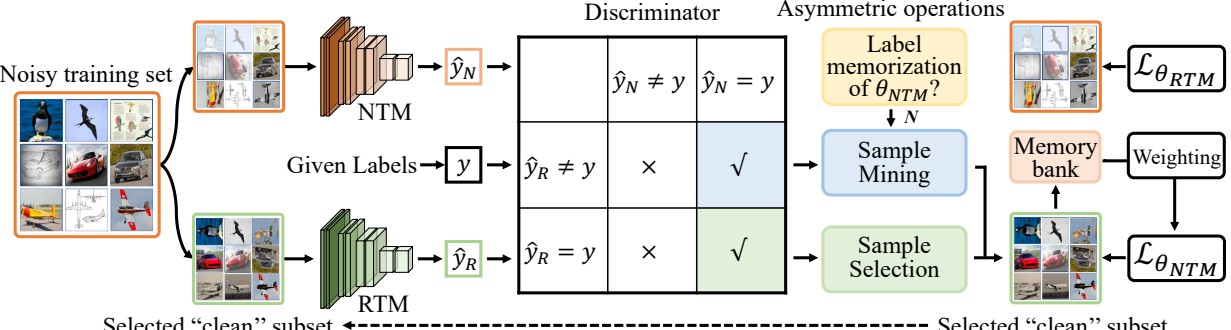

**Figure 2: The overall framework of our proposed ACT. We train two models simultaneously but employ robust (*i.e.*, training using selected clean data) and non-robust (*i.e.*, training on the entire noisy training set) training strategies separately. By revisiting the prediction results ($\hat{y}_R$ and $\hat{y}_N$) and the given labels $y$, we perform asymmetric sample selection ($\hat{y}_R = y$, $\hat{y}_N = y$) and sample mining ($\hat{y}_R! = y$, $\hat{y}_N = y$) before the non-robust model suffers from label memorization. Moreover, we maintain a memory bank to estimate the reliability of selected and mined "clean" samples. A dynamic sample re-weighting scheme is proposed based on the memory bank to integrate the reliability of "clean" data in the loss re-weighting process.**

In SCT methods, the two models have identical architectures but are initialized differently to acquire discrepant learning capabilities. For instance, Co-teaching [13] trains two networks simultaneously and selects small-loss data to teach the peer network during training. JoCoR [45] maintains two networks, training them with a joint loss to make their predictions converge. Co-learning [40] proposes to train a shared feature encoder with two distinctive prediction heads, maximizing their agreement in the latent space. Co-LDL [38] simultaneously trains two models and lets them communicate useful knowledge by selecting low-loss and high-loss samples for each other. However, our study posits that the additional information gain introduced by the SCT strategy is constrained, as the disparities between the two models primarily arise from random initialization. The dual models will eventually converge, leading to a diminution of their effectiveness in addressing noisy labels.

## 3 METHODS

### 3.1 Problem Statement

Considering a classification problem with $C$ classes, let us suppose that $\mathcal{X} \subset \mathcal{R}^d$ is the input space and $\mathcal{Y} = \{0, 1\}^C$ is the given label space (in a one-hot manner). We denote $D = \{(x, y) | x \in \mathcal{X}, y \in \mathcal{Y}\}$ as the training set, which is obtained from the joint distribution over $\mathcal{X} \times \mathcal{Y}$. For noisy label learning, the given label $y \in \mathcal{Y}$ is potentially "incorrect" and we use $y^*$ to represent the ground-truth label of the sample $x$. In conventional supervised learning, the DNN learns a mapping function $\mathcal{F} : \mathcal{X} \to \mathcal{Y}$ on the training set $D$ and optimizes the network parameters $\theta$ using the following cross-entropy loss:

$$\mathcal{L} = -\frac{1}{|D|} \sum_{(x,y) \in D} y \log(\mathcal{F}(x, \theta)). \tag{1}$$

The goal is to obtain optimal parameters $\theta^*$ by minimizing the empirical risk $\mathcal{R}_\mathcal{L}(\mathcal{F})$ subjected to network parameters as follows:

$$\theta^* = \arg\min_\theta \mathcal{R}_\mathcal{L}(\mathcal{F}(\cdot; \theta)). \tag{2}$$

Given the remarkable fitting capability of DNNs [62], optimization of network parameters using noisy labels within the conventional supervised learning framework can potentially steer the model toward an undesirable direction. Therefore, it is imperative to establish a solution capable of effectively addressing noisy labels.

### 3.2 Asymmetric Co-training

SCT has been demonstrated effective in learning with noisy labels, particularly in sample selection-based methods [13, 21, 28, 40, 45, 56, 61]. Resorting to the simultaneously trained dual networks, SCT effectively harnesses their diverse learning capabilities to promote model robustness in a mutual-reinforced manner. However, the two models in SCT are destined to converge due to the identical network architecture and the homogeneous training process, eventually vanishing the information gains obtained from symmetric training.

To this end, we propose an **A**symmetric **Co**-**T**raining (**ACT**) method, aiming to continuously enhance model robustness against noisy labels through asymmetric learning. In contrast to SCT, where both models adhere to the same training process, our ACT simultaneously trains two networks (*i.e.*, RTM and NTM) with identical architectures but employs distinct training strategies. RTM (*i.e.*, $\theta_{RTM}$) adopts a robust training strategy during network optimization, while NTM (*i.e.*, $\theta_{NTM}$) is trained with the entire training set following the conventional supervised learning process. As such, an asymmetric co-training framework is accordingly established. Specifically, to facilitate the robustness against noisy labels, we perform loss back-propagation only on a selected "clean" subset $D_c \subseteq D$ when training RTM. Its loss function is as follows:

$$\mathcal{L}_{\theta_{RTM}} = -\frac{1}{|D_c|} \sum_{(x,y) \in D_c} y \log(\mathcal{F}(x, \theta_{RTM})). \tag{3}$$

For the NTM, we follow the conventional supervised learning procedure, conducting training on the entire training set $D$. Its loss

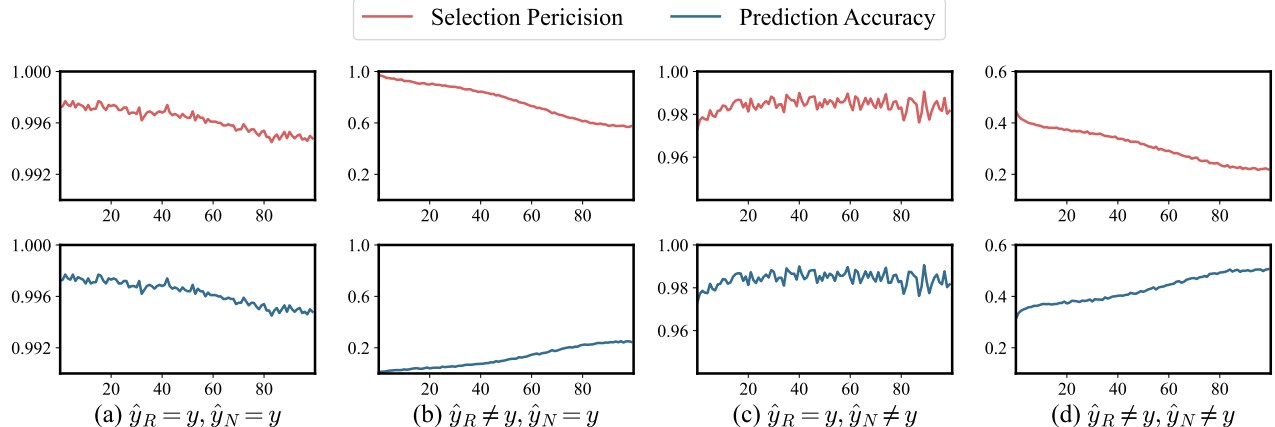

**Figure 3: Selection precision and prediction accuracy of samples selected by different criteria on CIFAR100N-Sym-20%.**

function is as follows:

$$\mathcal{L}_{\theta_{NTM}} = -\frac{1}{|D|} \sum_{(x,y) \in D} y \log(\mathcal{F}(x, \theta_{NTM})). \qquad (4)$$

Accordingly, given an input sample $x$, we can derive the prediction results produced by the two models as:

$$\hat{y}_R = \arg\max_{c=1,...,C} p(x, \theta_{RTM})^c, \quad \hat{y}_N = \arg\max_{c=1,...,C} p(x, \theta_{NTM})^c. \quad (5)$$

$p(x, \theta_{RTM})^c$ and $p(x, \theta_{NTM})^c$ represent the prediction probabilities of the sample $x$ on the category $c$ by models $\theta_{RTM}$ and $\theta_{NTM}$.

Obviously, the NTM is predetermined to overfit noisy samples and yield degenerated performance. However, our design enables the NTM to complement the RTM with knowledge learned from a different perspective. Specifically, based on our asymmetric training design, the RTM consistently engages in robust learning from clean samples, whereas the NTM progressively fits all samples (including noisy ones) as a result of label memorization. RTM and NTM tend to exhibit agreement when learning clean samples (*i.e.*, robust learning) but disagreement when learning noisy samples (*i.e.*, label memorization). Consequently, we argue that our asymmetric training can provide more unique insights for selecting clean samples compared to existing SCT methods.

### 3.3 Asymmetric Sample Selection and Mining

Existing SCT methods have investigated both agreement-based [13, 45] and disagreement-based [28, 61] sample selection strategies for addressing noisy labels. However, their reliabilities are prone to be compromised due to the converging behavior of SCT models. Especially in the later training stage, SCT models tend to produce consentaneous predictions even when confronted with noisy data. This diminishes their precision in selecting clean samples, thereby leading to degraded model performance. Inspired by existing agreement-based and disagreement-based sample selection methods, Our ACT revisits the relationship between predictions and given labels. By employing our asymmetric training design, we obtain insights that support us in devising better sample selection

criteria, aiding in the selection and mining of more valuable and reliable clean samples for RTM.

Specifically, we find the relationships between predictions of models (*i.e.*, $\hat{y}_R$ and $\hat{y}_N$) and given labels $y$ can be categorized into four situations: (1) $\hat{y}_R = y$ and $\hat{y}_N = y$, (2) $\hat{y}_R \neq y$ but $\hat{y}_N = y$, (3) $\hat{y}_R = y$ but $\hat{y}_N \neq y$, and (4) $\hat{y}_R \neq y$ and $\hat{y}_N \neq y$. As depicted in Fig. 3, we compare the selection precision and the prediction accuracy of corresponding samples w.r.t. their ground truth by conducting experiments on a synthetically noisy dataset. Fig. 3 (a) demonstrates the high selection precision (approaching 0.998) in situation (1). As the samples selected by situation (1) exhibit agreement between given labels and model predictions, it ensures the high accuracy of RTM predictions. Inspired by the results in Fig. 3 (a), we introduce a new criterion to select clean samples for RTM as follows:

CRITERION 1. *A sample $x$ is deemed clean if its predicted results of RTM and NTM are consistent and aligned with its given label $y$ (i.e., $\hat{y}_R = \hat{y}_N = y$).*

Therefore, the clean subset $D_c$ in our ACT that we select to participate in the training of $\theta_{RTM}$ and the corresponding noisy subset is defined as follows:

$$D_c = \{(x,y) | (x,y) \in D, \hat{y}_R = \hat{y}_N = y\}, \quad D_n = D - D_c. \quad (6)$$

Fig. 3 (b), (c), and (d) depict the selection precision and prediction accuracy for the latter three situations, where the predictions of models and given labels exhibit disagreement. In Fig. 3 (b), we observe the selection begins with high precision but exhibits a notable decreasing trend. Meanwhile, the prediction accuracy of $\theta_{RTM}$ is consistently low. Essentially, this case has the potential to mine additional clean samples that $\theta_{RTM}$ has not yet learned (*i.e.*, $\hat{y}_R \neq y$). In Fig. 3 (c), both the selection precision and the prediction accuracy are consistently high. This indicates that $\theta_{RTM}$ adeptly fits this subset of samples, meaning that little additional information can be unearthed from this subset. Results in Fig. 3 (d) demonstrate that this portion of data is not conducive to the robustness of $\theta_{RTM}$.

Indeed, the samples identified in scenario (2) hold greater importance for mining additional valuable clean samples to enhance the robust training of the RTM before the NTM starts to suffer from

label memorization. Since $\theta_{NTM}$ will memorize labels to fit noisy samples in the later training stage, the training accuracy of the two models gradually deviates. Accordingly, we design the following self-adaptive metric to measure the extent of label memorization for $\theta_{NTM}$:

$$\mathcal{T} = \frac{Acc(\theta_{NTM}) - Acc(\theta_{RTM})}{Acc(\theta_{RTM})}, \quad Acc(\theta) = \frac{1}{|D|} \sum_{(x,y) \in D} y = \hat{y}_\theta. \tag{7}$$

Inspired by the findings from Fig. 3 (b) and $\mathcal{T}$, we additionally introduce a novel criterion for mining more valuable clean samples:

CRITERION 2. *A sample $x$ will be mined as a clean sample if its given label $y$ does not match the prediction of RTM, yet aligns with that of NTM (i.e., $\hat{y}_R \neq y, \hat{y}_N = y$) before NTM starts to suffer from label memorization (i.e., $\mathcal{T} \leq \tau$).*

Once the condition $\mathcal{T} > \tau$ is triggered, samples selected by $\hat{y}_R \neq y, \hat{y}_N = y$ are no longer reliable and thus should be neglected from the training of RTM. Formally, the subset of selected and mined clean samples at the $K$-th epoch is defined as:

$$D'_c = D_c \cup \{(x,y) \in D_n \mid \hat{y}_R \neq y, \hat{y}_N = y, \mathcal{T} \leq \tau\}. \tag{8}$$

### 3.4 Dynamic Sample Re-weighting

Some previous works have revealed that determining the cleanness of samples solely based on the current model predictions could bring potential risks in data reliability. The challenge arises from the inevitable fluctuations in model training, making it difficult to prevent a few noisy samples from being leaked into $D'_c$, especially in scenarios with high noise rates.

To guarantee the efficacy of our ACT method, we further propose a dynamic sample re-weighting approach to foster the reliability of the selected and mined "clean" samples in $D'_c$. Specifically, we introduce a memory bank ($\mathcal{M}$) to store the selection results of all samples throughout the training process as follows:

$$\mathcal{M}^K(x) = \begin{cases} \mathcal{M}^{K-1}(x) + 1, if (x,y) \in D'_c \\ \mathcal{M}^{K-1}(x) + 0, if (x,y) \in D - D'_c \end{cases}. \tag{9}$$

$\mathcal{M}^K(x)$ denotes the number of epochs that $x$ falls into $D'_c$ at the $K$-th epoch. ($\mathcal{M}^0(x) = 0, 0 \leq \mathcal{M}^K(x) \leq K$.) Subsequently, we leverage the stored value of $\mathcal{M}$ for each clean sample as a weight coefficient when training $\theta_{RTM}$. The loss $\mathcal{L}_{\theta_{RTM}}$ used for the RTM can be re-write as:

$$\mathcal{L}_{\theta_{RTM}} = -\frac{1}{|D'_c|} \sum_{(x,y) \in D'_c} \frac{\mathcal{M}(x)}{K} y \log(\mathcal{F}(x, \theta_{RTM})), \tag{10}$$

in which $K$ denotes the $K$-th epoch in the training process.

Notably, existing sample selection methods often require dataset-dependent prior knowledge [13, 45] (*e.g.*, a pre-defined drop rate or threshold). This nature makes it challenging to swiftly adapt them to different real-world scenarios. In contrast, our ACT employs a data-driven, self-adaptive sample selection strategy, rendering it free from dataset-dependent priors. Thus, it is more suitable for real-world applications. Moreover, by incorporating the reliability of sample selection and mining into loss re-weighting, the risk of overfitting to noisy labels is further mitigated, resulting in improved model performance.

---

**Algorithm 1** Our proposed algorithm

**Input:** The training set $D$, the robust and non-robust networks $\theta_{RTM}$ and $\theta_{NTM}$, warm-up epochs $E_w$, total epochs $E_{total}$, batch size $bs$.

1: **for** $epoch = 1, 2, \ldots, E_{total}$ **do**
2:    **if** $epoch \leq E_w$ **then**
3:      **for** $iteration = 1, 2, \ldots$ **do**
4:        Fetch a mini-batch $B = \{(x_i, y_i)\}^{bs}$ from $D$;
5:        Calculate $\mathcal{L}_{\theta_{RTM}} = -\sum_{(x,y) \in B} y \log F(x, \theta_{RTM})$;
6:        Calculate $\mathcal{L}_{\theta_{NTM}} = -\sum_{(x,y) \in B} y \log F(x, \theta_{NTM})$;
7:        Update $\theta_{RTM}, \theta_{NTM}$ by optimizing $\mathcal{L}_{\theta_{RTM}}, \mathcal{L}_{\theta_{NTM}}$.
8:      **end for**
9:    **end if**
10:    **if** $E_w < epoch \leq E_{total}$ **then**
11:      **for** $iteration = 1, 2, \ldots$ **do**
12:        Select "clean" samples using Eq. (6);
13:        Mine more "clean" samples using Eq. (8);
14:        Re-weight samples in $D'_c$ using Eq. (9);
15:        Calculate $\mathcal{L}_{\theta_{RTM}}$ and $\mathcal{L}_{\theta_{NTM}}$ using Eqs. (11) and (4);
16:        Update $\theta_{RTM}, \theta_{NTM}$ by optimizing $\mathcal{L}_{\theta_{RTM}}, \mathcal{L}_{\theta_{NTM}}$.
17:      **end for**
18:    **end if**
19: **end for**

**Output:** The updated robust network $\theta_{RTM}$.

---

### 3.5 The Overall Framework

In summary, we introduce a novel asymmetric co-training approach to alleviate the harmful effects of noisy labels. We simultaneously train two models with identical architectures following different training processes. The RTM is trained with a selected clean subset, while the NTM is trained using the entire noisy training set. We introduce two novel criteria to select and mine clean samples more precisely. A metric is developed to evaluate the degree of label memorization for the NTM, enabling our method to perform mining only before the NTM starts to memorize noisy labels. Moreover, we propose a dynamic sample re-weighting strategy, incorporating the reliability of sample selection and mining to further boost the model performance. The overall learning procedure of our ACT is illustrated in Fig. 2 and Algorithm 1. In practice, we follow [39, 56] and further employ a consistency regularization loss for optimizing $\theta_{RTM}$. Our final objective loss function for RTM is as follows:

$$\mathcal{L}_{\theta_{RTM}} = \mathcal{L}_{\theta_{RTM}} + \lambda \mathcal{L}_{REG}, \tag{11}$$

where $\lambda$ is the weighting factor. $\mathcal{L}_{REG}$ denotes the consistency regularization (CR) loss, which encourages prediction consistency between weakly-augmented ($A_W$) and strongly-augmented ($A_S$) views of the input samples:

$$\mathcal{L}_{REG} = -\frac{1}{|D|} \sum_{(x,y) \in D} y_A \log(\mathcal{F}(A_S(x), \theta_{RTM})), \tag{12}$$

in which

$$y_A = p(A_W(x), \theta_{RTM}). \tag{13}$$

**Table 1: Average test accuracy (%) on CIFAR100N and CIFAR80N over the last ten epochs. Experiments are conducted under various noise conditions ("Sym" and "Asym" denote the symmetric and asymmetric label noise, respectively). † means we re-implement the method using its open-sourced code and default hyper-parameters.**

| Methods | Publication | CIFAR100N | | | CIFAR80N | | |
|---|---|---|---|---|---|---|---|
| | | Sym-20% | Sym-80% | Asym-40% | Sym-20% | Sym-80% | Asym-40% |
| Standard | - | 35.14 | 4.41 | 27.29 | 29.37 | 4.20 | 22.25 |
| Decoupling [28] | NeurIPS 2017 | 33.10 | 3.89 | 26.11 | 43.49 | 10.1 | 33.74 |
| Co-teaching [13] | NeurIPS 2018 | 43.73 | 15.15 | 28.35 | 60.38 | 16.59 | 42.42 |
| Co-teaching+ [61] | ICML 2019 | 49.27 | 13.44 | 33.62 | 53.97 | 12.29 | 43.01 |
| JoCoR [45] | CVPR 2020 | 53.01 | 15.49 | 32.70 | 59.99 | 12.85 | 39.37 |
| DivideMix [21] | ICLR 2020 | 57.76 | 28.98 | 43.75 | 57.47 | 21.18 | 37.47 |
| Jo-SRC [56] | CVPR 2021 | 58.15 | 23.80 | 38.52 | 65.83 | 29.76 | 53.03 |
| Co-LDL [38] | TMM 2022 | 59.73 | 25.12 | 52.28 | 58.81 | 24.22 | 50.69 |
| UNICON† [16] | CVPR 2022 | 55.10 | 31.49 | 49.90 | 54.50 | 36.75 | 51.50 |
| SOP† [26] | ICML 2022 | 58.63 | 34.23 | 49.87 | 60.17 | 34.05 | 53.34 |
| AGCE† [66] | TPAMI 2023 | 59.38 | 27.41 | 43.04 | 60.24 | 25.39 | 44.06 |
| DISC† [23] | CVPR 2023 | 60.28 | 33.90 | 50.56 | 50.33 | 38.23 | 47.63 |
| ANL† [57] | NeurIPS 2023 | 60.20 | 23.39 | 44.15 | 61.35 | 20.74 | 47.31 |
| NPN† [35] | AAAI 2024 | 62.76 | 31.69 | 57.11 | 63.78 | 25.25 | 58.50 |
| **Ours** | - | **65.51** | **40.74** | **63.48** | **67.09** | **38.58** | **64.40** |

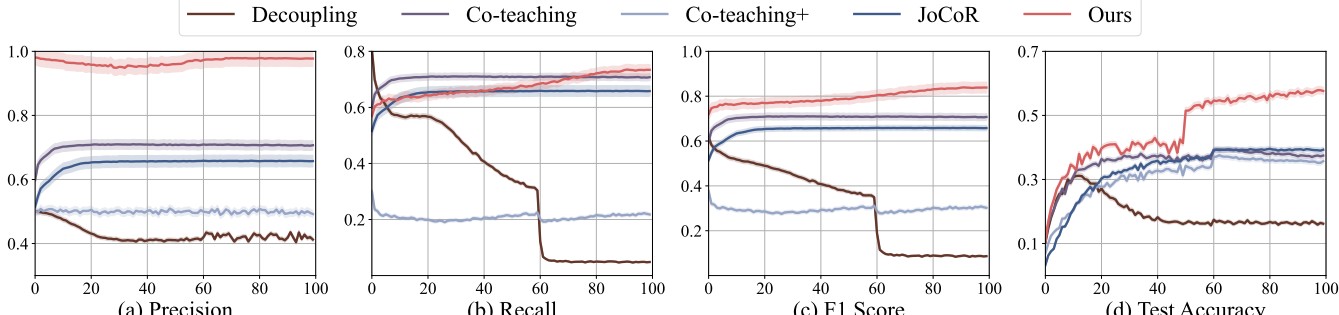

Figure 4: The comparison between SOTA methods and our ACT on precision, recall, F1 score, and test accuracy *vs.* epochs. Experiments are conducted on CIFAR100N with Sym-50%.

## 4  EXPERIMENTS

In this section, we first evaluate the effectiveness of ACT on various synthetic datasets. Then, we perform experiments on real-world benchmark datasets. Finally, we conduct ablation studies to investigate each ingredient in ACT. Further studies, such as the analysis of hyper-parameters, are provided in our supplementary material.

### 4.1  Experiment Setup

**Synthetic Datasets:** Following [56], we evaluate our ACT approach on two synthetic datasets (*i.e.*, CIFAR100N and CIFAR80N). CIFAR100N and CIFAR80N originate from CIFAR100 [17]. They are created to simulate closed-set and open-set noisy scenarios, respectively. Adhering to [56], we primarily study two types of synthetic label noise: symmetric (Sym.) and asymmetric (Asym.).

**Real-world Datasets:** Web-Aircraft, Web-Bird, and Web-Car [37] are three real-world noisy datasets whose training images are

crawled from web image search engines. In comparison to synthetic datasets, they present more significant challenges due to their unpredictable noise patterns. Moreover, it has been revealed that they contain both closed-set and open-set noise. Food-101N [18] is another benchmark dataset containing 101 food categories. It comprises around 310k noisy training images. The noise rate and structure are both unknown.

**Implementation Details:** We follow [56] to conduct experiments on synthetic datasets using a seven-layer CNN network as the backbone of our RTM and NTM. Accordingly, models are trained using SGD with a momentum of 0.9 for 150 epochs (including 50 warm-up epochs). To further promote the asymmetricity between the two models, we set the learning rates for the RTM and NTM as 0.01 and 0.08, respectively. The batch size is 128, and the learning rates decay in a cosine annealing manner. When experimenting on real-world datasets, we leverage ResNet50 [14] pre-trained on

**Table 2: The comparison with SOTA approaches in test accuracy (%) on real-world noisy datasets: Web-Aircraft, Web-Bird, Web-Car. $^\dagger$ means we re-implement the method using its open-sourced code and default hyper-parameters.**

| Methods | Publication | Backbone | Performances(%) | | | |
|---|---|---|---|---|---|---|
| | | | Web-Aircraft | Web-Bird | Web-Car | Average |
| Standard | - | ResNet50 | 60.80 | 64.40 | 60.60 | 61.93 |
| Decoupling [28] | NeurIPS 2017 | ResNet50 | 75.91 | 71.61 | 79.41 | 75.64 |
| Co-teaching [13] | NeurIPS 2018 | ResNet50 | 79.54 | 76.68 | 84.95 | 80.39 |
| Co-teaching+ [61] | ICML 2019 | ResNet50 | 74.80 | 70.12 | 76.77 | 73.90 |
| PENCIL [58] | CVPR 2019 | ResNet50 | 78.82 | 75.09 | 81.68 | 78.53 |
| JoCoR [45] | CVPR 2020 | ResNet50 | 80.11 | 79.19 | 85.10 | 81.47 |
| AFM [32] | ECCV 2020 | ResNet50 | 81.04 | 76.35 | 83.48 | 80.29 |
| DivideMix [21] | ICLR 2020 | ResNet50 | 82.48 | 74.40 | 84.27 | 80.38 |
| Jo-SRC [56] | CVPR 2021 | ResNet50 | 82.73 | 81.22 | 88.13 | 84.03 |
| Co-LDL [38] | TMM 2022 | ResNet50 | 81.97 | 80.11 | 86.95 | 83.01 |
| UNICON $^\dagger$ [16] | CVPR 2022 | ResNet50 | 85.18 | 81.20 | 88.15 | 84.84 |
| SOP$^\dagger$ [26] | ICML 2022 | ResNet50 | 84.06 | 79.40 | 85.71 | 83.06 |
| AGCE$^\dagger$ [66] | TPAMI 2023 | ResNet50 | 84.22 | 75.60 | 85.16 | 81.66 |
| DISC$^\dagger$ [23] | CVPR 2023 | ResNet50 | 85.27 | 81.08 | 88.31 | 84.89 |
| ANL$^\dagger$ [57] | NeurIPS 2023 | ResNet50 | 81.78 | 79.46 | 86.47 | 82.57 |
| NPN$^\dagger$ [35] | AAAI 2024 | ResNet50 | 83.65 | 79.36 | 85.46 | 82.82 |
| **Ours** | - | ResNet50 | **86.56** | **81.43** | **88.75** | **85.58** |

ImageNet-1K as our backbone. The batch size, the initial learning rate, and the weight decay are 16, 0.005, and 0.0005, respectively.
**Evaluation Metrics:** We adopt test accuracy as the primary metric to assess our model performance. Moreover, to enable a more comprehensive analysis, we additionally evaluate the results of sample selection by using the precision, recall, and F1 score metrics. Our reported performances are averaged results of five repeated runs.
**Baselines:** For synthetic datasets, we compare our ACT with following SOTA methods: Decoupling [28], Co-teaching [13], Co-teaching+ [61], JoCoR [45], DivideMix [21], Jo-SRC [56], Co-LDL [38], UNICON [16], SOP [26], AGCE [66], DISC [23], ANL [57] and NPN [35]. For real-world datasets, we additionally compare ACT with other competing methods (*e.g.*, PENCIL [58], AFM [32], PLC [64] and DivideMix+SNSCL [47]). Moreover, we perform conventional training using the entire noisy dataset as a baseline (denoted as Standard). Results of SOTA methods in Tables 1, 2 and 3 are mainly obtained from [56], [38] and [40].

## 4.2 Evaluation on Synthetic Datasets

Table 1 presents the comparison results on the synthetic datasets (*i.e.*, CIFAR100N and CIFAR80N) under various noise types (*i.e.*, symmetric and asymmetric) and noise rates (*i.e.*, 20%, 40% and 80%). Observing Table 1, we find it is evident that our ACT consistently outperforms all competing methods in various noisy conditions on these synthetic noisy datasets. Especially on CIFAR100N, the performances of our ACT excel existing approaches by notable margins (*i.e.*, 2.75%↑ on Sym-20%, 6.51%↑ on Sym-80%, and 6.37%↑ on Asym-40%), verifying the effectiveness of our method in coping with various closed-set noisy labels. Compared to CIFAR100N, CIFAR80N is undoubtedly more challenging since it is generated to mimic real-world cases where closed-set and open-set noisy labels simultaneously exist. Our ACT remains the top performer when

**Table 3: The comparison with SOTA approaches in test accuracy (%) on Food101N.**

| Methods | Publication | Backbone | Acc (%) |
|---|---|---|---|
| Standard | - | ResNet50 | 84.50 |
| Decoupling [28] | NeurIPS 2017 | ResNet50 | 85.53 |
| Co-teaching [13] | NeurIPS 2018 | ResNet50 | 61.91 |
| Co-teaching+ [61] | ICML 2019 | ResNet50 | 81.61 |
| JoCoR [45] | CVPR 2020 | ResNet50 | 77.94 |
| DivideMix [21] | ICLR 2020 | ResNet50 | 85.88 |
| Jo-SRC [45] | CVPR 2021 | ResNet50 | 86.66 |
| PLC [64] | ICML 2021 | ResNet50 | 85.28 |
| SNSCL [47] | CVPR 2023 | ResNet50 | 86.40 |
| **Ours** | - | ResNet50 | **86.81** |

compared with competing approaches on CIFAR80N. Although our method only achieves 0.35% performance improvement compared to the second-best counterpart (*i.e.*, DISC [23]) on CIFAR80N (Sym-80%), our ACT obtains remarkable performance gains in the other two cases (*i.e.*, 3.31%↑ on Sym-20% and 5.90%↑ on Asym-40%). This substantiates the efficacy of our proposed ACT method in adeptly tackling diverse challenging noisy labels.

To further demonstrate the efficacy of our ACT, we additionally investigate the performance of our asymmetric sample selection and mining by performing a comparison of sample identification results with existing SCT methods (*i.e.*, Decoupling, Co-teaching, Co-teaching+, and JoCoR), using the precision, recall, and F1 score metrics. Fig. 4 shows the comparison results on CIFAR100N with Sym-50% label noise. From Fig. 4 (a), it is evident that the selection precision of our ACT is significantly higher than that of other SCT methods. This observation suggests that the clean samples

**Table 4: Effects of different modules in test accuracy (%) on CIFAR100N and CIFAR80N under various noise conditions.**

| # | Model | CIFAR100N | | | CIFAR80N | | |
|---|-------|-----------|-----------|-----------|-----------|-----------|-----------|
| | | Sym-20% | Sym-80% | Asym-40% | Sym-20% | Sym-80% | Asym-40% |
| 1 | Standard | 35.14 | 4.41 | 27.29 | 29.37 | 4.20 | 22.25 |
| 2 | Standard+ASS | 59.30 | 28.59 | 41.69 | 59.90 | 26.70 | 42.08 |
| 3 | Standard+ASS+ASM | 63.25 | 33.91 | 54.82 | 64.66 | 32.81 | 56.66 |
| 4 | Standard+ASS+DSRW | 61.44 | 29.88 | 42.81 | 61.75 | 28.48 | 56.03 |
| 5 | Standard+ASS+ASM+DSRW | 63.66 | 35.89 | 59.77 | 65.46 | 34.42 | 60.41 |
| 6 | Standard+ASS+ASM+DSRW+CR | 65.51 | 40.74 | 63.48 | 67.09 | 38.58 | 64.40 |

discovered by our method are highly reliable. While the recall of our ACT starts at a relatively low level (which is the cost of ensuring the reliability of selected clean samples), it eventually surpasses its SCT counterparts, as shown in Fig. 4 (b). This demonstrates the effectiveness and robustness of our method in identifying clean samples. Consequently, the F1 score of ACT consistently excels all competing methods throughout the entire training process, as shown in Fig. 4 (c). Lastly, Fig. 4 (d) further demonstrates the leading performance of our method in test accuracy during training.

## 4.3 Evaluation on Real-world Datasets

Table 2 shows the comparison result between our ACT and existing SOTA methods on three real-world datasets (*i.e.*, Web-Aircraft, Web-Bird, and Web-Car). These datasets contain at least 25% of unknown noisy labels and do not provide any label verification information, rendering them both practical and challenging. Table 2 shows that our ACT consistently outperforms these competing methods. Specifically, ACT achieves 86.56%, 81.43%, and 88.75% accuracy on Web-Aircraft, Web-Bird, and Web-Car, respectively, surpassing the second-best performer DISC [23] by 1.29%, 0.35%, and 0.44%. The average test accuracy outperforms DISC by 0.69%. In particular, compared with classic SCT methods (*i.e.*, Decopuling, Co-teaching, Co-teaching+, JoCoR, and Co-LDL), ACT achieves an evidently significant performance improvement. The results, as depicted in Table 2, provide evidence for the robustness and generalization ability of our ACT method in handling real-world noisy labels.

Table 3 presents the performance comparison with SOTA methods on the Food101N dataset. As shown in Table 3, ACT achieves the best score and outperforms the state-of-the-art SNSCL [47] by 0.41%, validating the effectiveness of our approach in dealing with large-scale, real-world noisy cases.

## 4.4 Ablation Studies

This section, as illustrated in Table 4, investigates the effectiveness and impact of each ingredient (ASS, ASM, DSRW, and CR) in our method through ablation studies. Standard represents the conventional forward training using the cross-entropy loss. ASS and ASM denote the asymmetric sample selection and mining in our ACT. DSRW indicates the dynamic sample re-weighting process. CR means consistency regularization.
**Effects of Asymmetric Sample Selection and Mining:** Existing SCT methods often face challenges of model convergence, limiting

the knowledge acquired from model interactions. In our framework, we introduce two criteria (*i.e.*, Criteria 1 and 2) and formulate the asymmetric sample selection (ASS) and mining (ASM) strategy based on the relationship between model predictions and given labels. From the second row (#2) of Table 4, we can observe a striking and consistent performance improvement when employing our proposed ASS module. This confirms the capability of ASS in selecting clean samples from the consensus between RTM and NTM. Moreover, as depicted in the third row (#3) of Table 4, we can find that our proposed ASM also achieves remarkable performance gains. This validates the effectiveness of ASM in mining more valuable clean samples from the discrepancy between the two models.
**Effects of Dynamic Sample Re-weighting:** Due to the lack of ground truth for noisy labels, the identified "clean" samples are never necessarily reliable. Therefore, in our ACT method, we propose a dynamic sample re-weighting (DSRW) module that incorporates the reliability of selected clean samples in the process of loss weighting. DSRW introduces a surrogate metric to measure the reliability of selected samples based on training history, determining their weights in loss back-propagation. This further boosts the robustness against label noise. Table 4 demonstrates that DSRW brings consistent benefit to the model performance.
**Effects of Consistency Regularization:** Our proposed ACT employs consistency regularization (CR) to pursue additional performance gains. Consistency regularization enables us to unearth more knowledge from samples, including those discarded "unclean" and "unmined" ones. Consequently, as shown in Table 4, our model performance is further promoted.

## 5 CONCLUSION

In this paper, we proposed an asymmetric co-training (ACT) method to address noisy labels. ACT trained two models (*i.e.*, RTM and NTM) simultaneously in an asymmetric manner, equipping them with distinctive capabilities. We accordingly introduced an asymmetric sample selection and mining strategy to reliably identify and mine valuable clean samples. We established a metric based on the divergence between RTM and NTM to quantify label memorization, thereby guiding our ACT on the optimal juncture to cease sample mining. Moreover, a dynamic sample re-weighting scheme was proposed to incorporate the reliability of selected samples in the loss re-weighting process. Comprehensive experiments and ablation studies on various noisy datasets substantiated the effectiveness and superiority of our approach.

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
