# OpenReview forum: "Enhancing Robustness in Learning with Noisy Labels: An Asymmetric Co-Training Approach"
_acmmm.org/ACMMM/2024/Conference — MM2024 Poster_

### Official Review · Reviewer_Cs1n · 2024-05-21

**Rating:** 4
**Confidence:** 3

**Summary:**

The paper addresses the problem of label noise in training datasets, which significantly degrades the performance of deep neural networks. Traditional symmetric co-training methods, where two models with identical architectures are trained together, often converge and become less effective in handling noisy labels. To mitigate this, the authors propose an Asymmetric Co-Training (ACT) approach. This method involves training two models: a Robustly Trained Model (RTM) using a clean subset of the data, and a Non-Robustly Trained Model (NTM) using the entire dataset. The ACT framework introduces novel criteria for sample selection and mining based on model agreement and disagreement, guiding the training process and dynamically re-weighting samples based on their reliability.

**Strengths:**

1.This paper introduces a novel asymmetric co-training algorithm. By selecting consistently agreed-upon examples, the algorithm avoids overfitting to noisy examples.

2.The paper is well-organized, and the selected criteria are reasonable, supported by validation experiments.

3.The method significantly improves performance on a synthetic dataset, demonstrating the effectiveness of the proposed approach.

**Limitations:**

Weaknesses & Suggestions

1.The motivation is unclear. In the introduction, the authors suggest that the success of SCT mainly stems from distinct initializations. However, the success of ACT also heavily depends on distinct initializations to ensure the models are different enough to select clean examples. This similarity weakens the paper’s motivation.

2.The writing in Section 3.3 lacks clarity, with two main issues: 1) Figure 3 is difficult to understand because the x-axis is not defined, nor is the calculation of the metric explained. 2) The presentation is ambiguous; for instance, it's unclear whether scenario (2) mentioned in line 461 refers to the same situation as in line 430. As this subsection is central to the paper, it needs clearer organization and explanation.

3.The sensitivity analysis of key parameters is missing or incomplete. The number of warm-up epochs $E_w$ affects the training process: if too large, NTM starts to overfit noisy data, while if too small, the model agreement resembles a random result. The choice of this parameter also impacts the selection of $\tau$, making its analysis crucial. For $\tau$, sensitivity analysis in the Appendix indicates that most optimal values occur at $\tau=0.25$. Exploring what happens with $\tau < 0.25$ and its implications could be valuable. Moreover, the claim that the model is robust to changes in $\tau$ is questionable, as performance drops significantly in Sym-80%.

4.Compared to synthetic datasets, the proposed method shows limited improvement on real-world datasets. Exploring hyper-parameter selection might help extend the success from synthetic to real-world datasets.

5.It would be helpful to highlight the best competitor in the table.

Typos

1.In Eq. (1), it would be clearer if $\theta$ were denoted as a subscript rather than an input of F, given that it is represented as a mapping from X to Y.

2.In statement (2), if it is characterized as empirical risk, then it should be associated with D.  Additionally, here you use a semicolon in F.

3.In the Appendix, line 153, change $\lambda$ and $\tau$.

Although this paper has significant room for improvement, its novelty and potential are evident. Therefore, I rate it as borderline and am willing to raise my score following the rebuttal period.

**Suitability:**

2

---

### Official Review · Reviewer_Ukqn · 2024-05-23

**Rating:** 4
**Confidence:** 4

**Summary:**

The paper suggests training two networks on different data to achieve asymmetric training. Among them, RTM trains on the selected clean samples, while NTM trains on the entire noisy training set. It also leverages historical training states and the consensus and disagreement between the two models to facilitate robust training.

**Strengths:**

[1] This paper presents everything very well. The layout of the figures and text is exquisite, the introduction of related work is detailed and thorough, and the presentation of the main ideas and experiments is also sufficiently satisfactory.
[2] This work is a simple but effective improvement to the strategy of co-teaching.

**Limitations:**

The experiments presented in this article are confusing. CIFAR-100N is the name of a natural noise dataset, but the authors use CIFAR100N in the text to refer to a synthetic noise dataset with a certain label noise rate. The authors should use a different term to describe this synthetic dataset to avoid confusion for readers. Moreover, I can confidently say that many methods used in Table 1 do not involve the evaluation baselines using CIFAR100N and CIFAR80N (perhaps the authors can point me to works that use these datasets to evaluate method performance. The authors claim they are following Jo-SRC, Jo-SRC claims it is following JoCoR, but JoCoR follows standard practices without using terms like 80N). CIFAR100N and CIFAR80N seem to be a very challenging synthetic version of CIFAR100. Under Sym-20% noise, SOP and DivideMix, two methods considered to have excellent performance in learning with label noise, can only achieve 50%+ performance. Table 2 does not appear to be a standard baseline test either. Notably, on Food101N, the authors' method achieves similar performance to DivideMix.
Although I agree that performance is a relatively unimportant factor in whether a proposed method should be recognized, as a reader, I expect a performance comparison on comparable and widely-recognized baselines to convince me of the effectiveness of the proposed method.

**Suitability:**

3

---

### Official Review · Reviewer_t5pJ · 2024-05-23

**Rating:** 2
**Confidence:** 4

**Summary:**

This paper proposes an Asymmetric Co-Training (ACT) method to enhance the robustness of deep neural networks (DNNs) when dealing with noisy labels. The ACT method involves training two models, the Robust Training Model (RTM) and the Noisy Training Model (NTM), using different strategies: RTM is trained with a subset of clean samples, while NTM is trained with the entire dataset. The method introduces novel sample selection criteria based on consistency and discrepancy between model predictions and labels, as well as a dynamic sample re-weighting mechanism that utilizes historical information. The effectiveness of the ACT method is demonstrated through experiments on both synthetic datasets (CIFAR100N and CIFAR80N) and real-world datasets (Web-Aircraft, Web-Bird, Web-Car, and Food-101N).

**Strengths:**

The paper proposes a novel Asymmetric Co-Training (ACT) method to enhance the robustness of models when dealing with noisy labels. The Robust Training Model (RTM) and the Noisy Training Model (NTM) are trained using different strategies: RTM with a subset of clean samples and NTM with the entire dataset. Additionally, the paper introduces new sample selection criteria based on the consistency and discrepancy of model predictions, as well as a dynamic sample re-weighting mechanism to improve the reliability of sample selection. These innovative strategies offer a new approach in the realm of co-training methods.

**Limitations:**

1. Lack of Theoretical Support

- No Theoretical Proof: Although the paper proposes a novel method, it lacks theoretical validation. There is insufficient theoretical analysis to support the effectiveness of the proposed Asymmetric Co-Training (ACT) method.
- Insufficient Experiment Robustness: The experiments were not repeated multiple times, and standard deviations were not reported, making it difficult to assess the robustness of the results.

2. Poor Experimental Performance

- Limited Improvement on Real Datasets: The experimental results on real-world datasets show limited improvement, failing to fully demonstrate the advantages of the method.

3. Issues with Dataset Selection
- Unconventional Datasets: The datasets used in the experiments are not commonly used in this field, such as CIFAR-10, SVHN, or Clothing1M. Notably, CIFAR-100N generally refers to another real-world dataset (as seen in this paper: https://arxiv.org/pdf/2110.12088v2).
- Limited Noise Types: The authors should consider using more diverse types of noise, including pairflip and instance-dependent noise, to more comprehensively validate the method’s effectiveness.

4. Threshold Determination
- Insufficient Assumptions: The paper does not provide sufficient assumptions for the self-adaptive metric used to determine when NTM fits the noise. Additionally, experiments should be added to validate the effectiveness, enhancing the credibility of the method.

**Suitability:**

2

---

### Meta-Review · Area_Chair_1bmL · 2024-06-29

**Recommendation:** Accept (Poster)
**Confidence:** 5

**Metareview:**

The paper, along with the assessments and the authors' rebuttal, has been thoroughly reviewed by the ACs. ACs also considered the confidential comments sent by the authors in the post-rebuttal discussions and their final decision. The authors introduce an asymmetric co-training method to tackle noisy labels in deep neural networks. While concerns, such as the lack of theoretical support, were raised initially, all reviewers have recommended acceptance after the rebuttal. The ACs concur with this decision and therefore recommend accepting the paper. Congratulations to the authors. Please carefully consider the reviewers' feedback and suggestions for revision, ensuring to address the promises made in the rebuttal.